# A Partial Carrier Phase Integer Ambiguity Fixing Algorithm for Combinatorial Optimization between Network RTK Reference Stations

**DOI:** 10.3390/s22010165

**Published:** 2021-12-27

**Authors:** Shouhua Wang, Zhiqi You, Xiyan Sun

**Affiliations:** 1College of Information and Communication, Guilin University of Electronic Technology, Guilin 541004, China; youzhiqi_mail@163.com (Z.Y.); sunxiyan1@163.com (X.S.); 2Key Laboratory of Cognitive Radio and Information Processing, School of Information and Communication, Guilin University of Electronic Technology, Ministry of Education, Guilin 541004, China; 3National Engineering Research Center for Satellite Navigation, Positioning and Position Service, Guilin 541004, China

**Keywords:** network RTK, ambiguity resolution, partial ambiguity fixation, robust extended Kalman filtering

## Abstract

In the face of a complex observation environment, the solution of the reference station of the ambiguity of network real-time kinematic (RTK) will be affected. The joint solution of multiple systems makes the ambiguity dimension increase steeply, which makes it difficult to estimate all the ambiguity. In addition, when receiving satellite observation signals in the environment with many occlusions, the received satellite observation values are prone to gross errors, resulting in obvious deviations in the solution. In this paper, a new network RTK fixation algorithm for partial ambiguity among the reference stations is proposed. It first estimates the floating-point ambiguity using the robust extended Kalman filtering (EKF) technique based on mean estimation, then finds the optimal ambiguity subset by the optimized partial ambiguity solving method. Finally, fixing the floating-point solution by the least-squares ambiguity decorrelation adjustment (LAMBDA) algorithm and the joint test of ratio (R-ratio) and bootstrapping success rate index solver. The experimental results indicate that the new method can significantly improve the fixation rate of ambiguity among network RTK reference stations and thus effectively improve the reliability of positioning results.

## 1. Introduction

China’s strong support for BeiDou satellite navigation and positioning technology and the continuous demand for positioning services require that conventional real-time kinematic (RTK) technology be more optimized both in terms of the solving accuracy of the fixed ambiguity solution, and baseline application distance between the monitoring station and the reference station. Therefore, the network RTK positioning technology based on the Continuously Operating Reference Station (CORS) system arose, compared to the conventional RTK technology, which possesses the advantages of broader applications, more accurate dynamic navigation and positioning, and more reliability. This technology uses multiple reference stations to form a reference station network and provides precise differential correction data (difference between the actual and measured values) for mobile stations within the coverage of the network, thus maximizing the observation baseline length. The accuracy of the ambiguity fixed solution for the network RTK inter-reference station will directly affect the reliability of the monitoring station [1,2]. When solving the network RTK inter-reference station ambiguity fixed solution timely, the observation value of the network RTK reference station is prone to gross error as the observation environment is subject to more occlusions resulting in poor satellite signals. This makes the parameter estimation results appear clear deviated or even lead to filter divergence, further affecting the overall results of network RTK precious positioning [3].

Ma et al. [4] employed the MW (Melbourne-Wubeena combination) combined observation equation to solve the whole-period ambiguity of the double-difference broad alley at the reference station and then used the integrated error interpolation method to solve the error correction number of the reference station and the mobile station to correct the ambiguity. However, this method is computationally intensive and insensitive to the baseline length. Li et al. [5] used the combined ionosphere-free observations for the determination of the whole-period ambiguity of the double-difference carrier phase between the reference stations, yet there are still some issues: (1) the problem of the success rate of ambiguity settlement solved by the initially rising satellite; (2) the problem of the success rate of ambiguity settlement when there is a large amount of circumferential hopping information in the observed signal. Gao et al. [6] proposed a fast solution based on a partial ambiguity fixing strategy method, which first shortens the ambiguity solution fixation time among reference stations by establishing an ionospheric model and then screening the optimal subset of the ambiguity set.

Reliable estimation of the ambiguity is a key issue in network RTK localization, whereas the joint solution of multiple systems makes the dimension of the whole circumference ambiguity increase steeply, resulting in a more difficult estimation of all ambiguities, especially prominent when the observation environment is poor [7]. Moreover, in a more complex observation environment, gross errors will appear in the observation values, which is the main factor limiting the fast and accurate solution of the ambiguity among network RTK reference stations [8,9,10]. Domestic and foreign researchers have demonstrated that fixing all ambiguities at once is unnecessary [11,12]. This paper first estimates the floating-point ambiguity using the robust extended Kalman filtering technique, then utilizes the optimized partial ambiguity solving method, and fixes the solution of the ambiguity by using the LAMBDA algorithm and the joint test of ratio (R-ratio) and bootstrapping success rate. The quality control of the ambiguity estimation process of the network RTK reference station is realized, and the fixed value of the whole-period fuzzy degree is constrained, and the next solving step is fed by the calendar element (representing different observation times), which accordingly achieves high accuracy in the network RTK positioning.

## 2. Materials and Methods

### 2.1. GNSS Network RTK Data Solution Model

Typically, the carrier phase observation equation and the pseudo-range observation equation can be established between the receiver and the satellite:(1)$$\rho_{r}^{i}=R_{r}^{i}+c\delta t_{r}-c\delta t^{i}+I_{r}^{i}+T_{r}^{i}+\xi_{\rho ,r}^{i}$$
(2)$$\varphi_{r}^{i}=\lambda^{-1}\left(R_{r}^{i}+c\left(\delta t_{r}-\delta t^{i}\right)-I_{r}^{i}+T_{r}^{i}\right)+N_{r}^{i}+\xi_{\varphi ,r}^{i}$$
where $\rho_{r}^{i}$ and $\varphi_{r}^{i}$ represent pseudo-range observations and carrier phase observations, respectively; $I_{r}^{i}$ is the measured ionospheric delay; $\xi_{\rho ,r}^{i}$ is the amount of pseudo-range observation noise; $\xi_{\varphi ,r}^{i}$ represents the amount of carrier phase observation noise; $\delta t_{r}$ represents the amount of receiver clock error; $T_{r}^{i}$ represents the tropospheric delay error; $R_{r}^{i}$ represents the distance between the receiver and the satellite; $\delta t^{i}$ is the satellite clock error; $N_{r}^{i}$ is the ambiguity that needs to be calculated; c is the speed of light. $R_{r}^{i}$ can be determined by the receiver coordinate $\left(x,y,z\right)$ and the satellite coordinate $\left(x^{i},y^{i},z^{i}\right)$ as:(3)$$R_{r}^{i}=\sqrt{{\left(x-x^{i}\right)}^{2}+{\left(y-y^{i}\right)}^{2}+{\left(z-z^{i}\right)}^{2}}$$

### 2.2. Solving for Ambiguity

Currently, the solution of ambiguity is usually fixed using the least-squares ambiguity decorrelation adjustment (LAMBDA), which is a search method based on the ambiguity domain using mainly the principle of integer least-squares estimation.

The integer least-squares principle can be written in the following form:(4)y=Aa+Bb+ξ
where y is the vector of double-difference observations, A is the coefficient matrix of the ambiguity parameter, B is the coefficient matrix of the non-ambiguity parameter, b is the vector of other non-ambiguity parameters, a is the ambiguity parameter vector (a is an integer), and ξ is the noise vector. 

When solving Equation (4) using the least-squares principle, the solution process is often referred to as integer least squares estimation since it contains both integer and real parameters to be solved, and in general, the solution has no analytic form. The integer least squares estimation is usually solved according to the following three processes [13], as shown in the processing flow illustrated in Figure 1.

In the first step, the floating-point solution of the ambiguity parameter and its covariance matrix are first calculated without considering the ambiguity as an integer:(5)$$\left[\begin{array}{c} \tilde{a}\\ \tilde{b} \end{array}\right],\left[\begin{array}{cc} Q_{\tilde{a}\tilde{a}} & Q_{\tilde{a}\tilde{b}}\\ Q_{\tilde{b}\tilde{a}} & Q_{\tilde{b}\tilde{b}} \end{array}\right]$$

Among them, the superscript “~” means floating point.

In the second step, after the floating-point solution vector $\tilde{a}$ and the integer vector a are obtained, the integer solution of the fuzzy degree is found by minimizing the distance between the floating-point solution vector and the integer vector and by searching for the fuzzy degree parameter a:(6)$$\underset{a}{\min}{\left(\tilde{a}-a\right)}^{T}Q_{\tilde{a}\tilde{a}}^{-1}\left(\tilde{a}-a\right)$$

In the third step, after obtaining the integer solutions of the whole-period ambiguity $\hat{a}$, the fixed solutions of the non-ambiguity parameters and their covariance matrices are obtained by the following two equations:(7)$$\overset{⌢}{b}=\tilde{b}-Q_{\tilde{b}\tilde{a}}Q_{\tilde{a}\tilde{a}}^{-1}\left(\tilde{a}-\overset{⌢}{a}\right)$$
(8)$$Q_{\overset{⌢}{b}\overset{⌢}{b}}=Q_{\tilde{b}\tilde{b}}-Q_{\tilde{b}\tilde{a}}Q_{\tilde{a}\tilde{a}}^{-1}Q_{\tilde{a}\tilde{b}}$$

The LAMBDA method specifies the following search space when performing the search in the second step [14,15,16]:(9)$${\left(\tilde{a}-a\right)}^{T}Q_{\tilde{a}\tilde{a}}^{-1}\left(\tilde{a}-a\right)<T$$
where T is the threshold, the integers in the above search space are all search objects, and a set of integer vectors can satisfy Equation (6).

The strong correlation between the fuzzy degrees leads to a relatively low efficiency of the LAMBDA algorithm search. Therefore, it is necessary to first reduce the correlation between the covariance matrix and the fuzzy degree parameters using a Z transformation [17], and then search them using the LAMBDA algorithm with the following procedure:(10)$$\tilde{z}=Z^{T}\tilde{a}$$
(11)$$Q_{\tilde{z}\tilde{z}}=Z^{T}Q_{\tilde{a}\tilde{a}}Z$$

The reduced correlation matrix Z should satisfy (1) full rank, not only forward transformable but also inverse transformable; (2) $\left|detZ\right|\;=\;1$; (3) integer property to ensure that the fixed ambiguity is still an integer after the inverse transformation.

At this point, Equation (9) accordingly, becomes:(12)$${\left(\tilde{z}-z\right)}^{T}Q_{\tilde{z}\tilde{z}}^{-1}\left(\tilde{z}-z\right)\;<T$$

Then the search is performed using a sequential least squares method based on the $LDL^{T}$ decomposition and the desired integer solution for the ambiguity is obtained.

### 2.3. Robust Extended Kalman Filter Algorithm

The solution of the standard Kalman filtering algorithm will satisfy the minimum unbiased variance only when the observation model and the state model are comparatively accurate, and the noise of the observation signal and the state noise satisfy Gaussian white noise, are uncorrelated with each other, and the mean value is zero. However, when the observation environment is in a more complex situation, it is usually difficult to meet the assumptions [18,19]. For example, when there is a gross error in the observation data, or there is a missed period jump, the observation information will be contaminated, which will lead to jumping points in the localization results, and even malfunction of the filter and the phenomenon of filter divergence; therefore, the accuracy, as well as the reliability of the results, cannot be guaranteed [20,21]. At this time, if the mean estimation based on the anti-divergence Kalman filter is applied for the solution, it detects the gross errors according to the observation residuals and downscales the observations with excessive residuals. This achieves the anti-divergence effect and then effectively suppresses the influence of the gross errors on the results, improves the reliability of the results, and guarantees its solution accuracy.

The difference-resistant EKF introduces robust estimation theory into the parity model, which ensures that the valuation is near-optimal when the actual model matches the prior hypothesis model, ensures the valuation is less affected by the coarse difference when the actual model slightly deviates from the prior hypothesis model, and ensures that the estimate will not change significantly when the actual model deviates from the hypothesis model by a large margin [22]. The more commonly used robust estimation is the robust mean estimation, and the anti-variance extended Kalman filter introduces the IGG III model from the robust M estimation into the extended Kalman filter processing, where the IGG III model divides the observed residuals into three segments, constructs the anti-variance equivalent weights according to the adaptive downscaling factor, $r_{i}$ and uses iterative computation to obtain the anti-variance solution [23].

The adaptive downscaling factor $r_{i}$ in IGG III model is:(13)$$r_{i}=\left\{\begin{array}{ccc} 1 & , & \left|{\bar{v}}_{i}\right|\leq k_{1}\\ \frac{k1}{\left|{\bar{v}}_{i}\right|}{\left(\frac{k_{2}-\left|{\bar{v}}_{i}\right|}{k_{2}-k_{1}}\right)}^{2} & , & k_{1}<\left|{\bar{v}}_{i}\right|<k_{2}\\ 0 & , & \left|{\bar{v}}_{i}\right|\geq k_{2} \end{array}\right.$$
where $k_{1}=1.5\sim 2.5$, $k_{2}=3.0\sim 5.0,\;\bar{v_{i}}$ are the standardized residuals of the observations, and the calculation formula is:(14)$${\bar{v}}_{i}=\frac{V_{i}}{std(V)}$$
where std() is the standard deviation function, and V is the vector of observed residuals.

After obtaining the adaptive downscaling factors, the equivalent variance and covariance of the observations can be obtained as follows:(15)$$\left\{\begin{array}{l} {\bar{Q}}_{i,i}=\frac{1}{r_{i,i}}Q_{i,i}\\ {\bar{Q}}_{j,j}=\frac{1}{r_{j,j}}Q_{j,j}\\ {\bar{Q}}_{i,j}=\frac{1}{\sqrt{r_{i,i}r_{j,j}}}Q_{i,j} \end{array}\right.$$
where $Q_{i,i}$ and $Q_{j,j}$ represent the variances of observations i and j, and their equivalent variances are ${\bar{Q}}_{i,i}$ and ${\bar{Q}}_{j,j}$, respectively. $Q_{i,j}$ represents the covariance of observations i and j, and its equivalent covariance is ${\bar{Q}}_{i,j}$.

After determining the equivalent variance-covariance array consisting of equivalent variance and covariance $\bar{Q}$, the measurement update process of the extended Kalman filter is obtained by replacing the measurement variance-covariance matrix in the extended Kalman filter with the equivalent array $\bar{Q}$, which can be expressed as follows:(16)$${\bar{K}}_{k}=P_{k}^{-}H_{}^{T}{\left(HP_{k}^{-}H_{}^{T}+\bar{Q}\right)}^{-1}$$
(17)$${\hat{X}}_{k}={\hat{X}}_{k}^{-}+{\bar{K}}_{k}\left(y_{k}-H\left({\hat{X}}_{k}^{-}\right)\right)$$
(18)$${\bar{P}}_{k}=\left(I-{\bar{K}}_{k}H\right)P_{k}^{-}$$
where ${\bar{K}}_{k}$ represents the Kalman gain matrix; $P_{k}^{-}$ represents the prediction covariance matrix; ${\hat{X}}_{k}$ represents the optimal estimation matrix; ${\hat{X}}_{k}^{-}$ represents the state prediction matrix; H represents the state observation matrix; I represents the identity matrix; $y_{k}$ represents the observations of the state matrix.

### 2.4. Improving the Partial Ambiguity Fixed Solution

Considering the increase in Global Navigation Satellite System (GNSS) and the exponential increase in the number of satellites, researchers at home and abroad have shown that fixing all ambiguity parameters is unnecessary, and the risk of ambiguity fixation failure can be reduced by selecting a partial subset of ambiguities for fixation under the condition that the number of participating satellites is sufficient. Teunissen [21] first proposed a partial ambiguity fixation strategy for the selection of the optimal subset and a fixation strategy for the selection of the optimal subset based on the bootstrapping success rate. Some scholars also proposed partial ambiguity resolution (PAR) algorithms based on satellite elevation angle, signal-to-noise ratio, continuous tracking ephemeris number, ambiguity dilution of precision (ADOP) value, ambiguity variance, etc., in the following. Now, based on the PAR idea, a new partial ambiguity fixation method is proposed for the network RTK reference station ambiguity solution, which improves the ambiguity fixation rate by selecting the optimal subset at the satellite level and ambiguity level, i.e., taking full advantage of the data from satellites with high elevation angles to participate in the floating-point solution, and shortening the convergence time compared with the search of all the observed data.

The basic idea of partial fuzzy degree fixation is to divide the fuzzy degree into two subsets based on some criterion, giving priority to fixing the more easily fixed subset [24]. Specifically, the ambiguity vector $\tilde{a}$ is divided into two subsets ${\left[{\tilde{a}}_{1}\;{\tilde{a}}_{2}\right]}^{T}$, then its variance-covariance matrix can be expressed accordingly as:(19)$$Q_{\tilde{a}\tilde{a}}=\left[\begin{array}{cc} Q_{{\tilde{a}}_{1}{\tilde{a}}_{1}} & Q_{{\tilde{a}}_{1}{\tilde{a}}_{2}}\\ Q_{{\tilde{a}}_{2}{\tilde{a}}_{1}} & Q_{{\tilde{a}}_{2}{\tilde{a}}_{2}} \end{array}\right]$$
where ${\tilde{a}}_{1}$ represents the selected ambiguity subset that is easier to fix, and ${\tilde{a}}_{2}$ represents the remaining ambiguity subset. When the subset ${\tilde{a}}_{1}$ is fixed by the LAMBDA method, the integer solution of ${\tilde{a}}_{1}$ can be utilized to modify ${\tilde{a}}_{2}$ and its variance-covariance matrix $Q_{{\tilde{a}}_{2}{\tilde{a}}_{2}}$ to obtain the integer solution of ${\tilde{a}}_{2}$ and its fixed variance-covariance matrix:(20)$$\left\{\begin{array}{l} {\overset{⌢}{a}}_{2}={\tilde{a}}_{2}-Q_{{\tilde{a}}_{2}{\tilde{a}}_{1}}Q_{{\tilde{a}}_{1}{\tilde{a}}_{1}}^{-1}\left({\tilde{a}}_{1}-{\overset{⌢}{a}}_{1}\right)\\ Q_{{\overset{⌢}{a}}_{2}{\overset{⌢}{a}}_{2}}=Q_{{\tilde{a}}_{2}{\tilde{a}}_{2}}-Q_{{\tilde{a}}_{2}{\tilde{a}}_{1}}Q_{{\tilde{a}}_{1}{\tilde{a}}_{1}}^{-1}Q_{{\tilde{a}}_{1}{\tilde{a}}_{2}} \end{array}\right.$$

The specific steps of this method are as follows:Firstly, we try to fix all ambiguities, i.e., without rejection filtering of the ambiguity set, the search of the floating-point ambiguity parameters obtained by Kalman filtering using LAMBDA algorithm directly, and the search results are tested jointly by R-ratio and bootstrapping success rate indicators, and if they pass the test, the fixation is considered successful; otherwise, we proceed to the next step.The ambiguity subsets are divided at the satellite level, and the total set of satellite azimuths (0°~360°) is divided into one subset as long as it is over 90°, and subsets are denoted as $Q_{1}$, $Q_{2}$, $Q_{3}$, and $Q_{4}$ separately, and one satellite is removed from the divided subset to ensure that the GDOP value of the set is the smallest after removing a satellite. Then, the ambiguity of all remaining satellite sets $H_{1}$ is fixed.If the second step fails to be fixed, the set $H_{1}$ is then optimally filtered at the fuzzy degree level, and the three fuzzy degrees with larger ADOP values $\alpha_{max1}$, $\alpha_{max2}$, $\alpha_{max3}$ are selected from the subset. Then the GDOP1 value of the remaining satellite constellations can be calculated after eliminating the $\alpha_{max1}$ and $\alpha_{max2}$ ambiguities. The GDOP2 value of the remaining satellite constellations can be calculated after eliminating the $\alpha_{max1}$ and $\alpha_{max3}$ ambiguities, and the GDOP3 value of the remaining satellite constellations can be calculated after eliminating the $\alpha_{max2}$ and $\alpha_{max3}$ ambiguities. Comparing the size of the three GDOP values and eliminating the two ambiguities with large GDOP values from the set $H_{1}$ to obtain the optimal subset.The LAMBDA algorithm is employed to search the filtered optimal fuzzy degree subset and perform a joint R-ratio and bootstrapping success rate metric test, and if the fixation is successful, calculate the final fixed solution; if the fixation fails, the floating-point solution is saved.


## 3. Results

In order to analyze the performance of the robust extended Kalman filter algorithm, this subsection analyzes and verifies its performance by means of measured data. The data were collected from a monitoring station in the deformation monitoring area of a mining area in Inner Mongolia for a total of 3600 calendar elements (time corresponding to data), and measured data are collected from monitoring Station 1 in the deformation monitoring area of a ship lock in Hengxian County, Guangxi for a total of 3600 calendar elements.

The set of data is solved using the robust EKF ($k_{1}$ take 2.0, $k_{2}$ take 3.0), and the results are shown in Figure 2, Figure 3, Figure 4 and Figure 5. It can be seen that there are almost no outliers in the solved results after using the robust EKF. From the Figure 3 and Figure 5, it can be seen that the ephemeris in which coarse differences appear in the conventional EKF are almost all free from coarse differences in the carrier phase double difference residuals after using the robust EKF, and nearly 95% of the coarse differences are suppressed or eliminated.

As can be seen from the residual plots and the solved result plots, the coarse difference values appear in both horizontal and elevation aspects when the conventional EKF algorithm is used for parameter estimation, which seriously affects the overall accuracy of the solved results and cannot meet the needs of practical applications. When the robust EKF algorithm is used for processing, the coarse differences of both sets of measured data are effectively suppressed during the observation period, and the results are smoother. In terms of the coarse differences, compared to the conventional EKF algorithm, a robust EKF algorithm greatly improves the accuracy.

It is clear from the above experiments that the robust extended Kalman filter demonstrates effectiveness in coarse difference processing and illustrates the need for quality control of parameter estimation.

The commonly used PAR algorithms for optimal subset selection strategies are usually based on satellite level, where satellites with poor accuracy are eliminated based on satellite elevation angle, signal-to-noise ratio, or the number of successive locking ephemerides. Compared with observation satellites with high elevation angles, low elevation angle observation satellites are more susceptible to interference factors, especially errors that cannot be weakened or eliminated by differential methods such as multi-path errors. 

In order to analyze and verify the effectiveness of the proposed algorithm, the measured data are selected for experimental validation. In this case, the monitoring area of Baise City, Guangxi Zhuang Autonomous Region, with a baseline distance of 75 km between reference stations, is selected, and the data is collected using a laboratory-developed BDS/GPS receiver with a total of 2735 calendar elements. The experimental data acquisition points are shown in Figure 6.

The full ambiguity resolution (FAR) algorithm, the conventional PAR algorithm, and the new algorithm proposed in this paper are used to solve this set of data, where the FAR algorithm only sets the cut-off angle of satellite elevation angle (set to 15° in this case), and satellites with elevation angles lower than this value do not participate in the construction of the double-difference equation. The conventional PAR algorithm uses a simple combination of satellite elevation angle and the number of consecutive locked calendar elements on the basis of the FAR algorithm to solve the fuzzy degree subset to filter and achieve partial ambiguity fixation. One of the relevant parameter settings is shown in the following Table 1.

The ADOP threshold is set to 0.14 when the number of ambiguities is within the [6,10] interval, 0.135 when the number of ambiguities is within the interval [11,15], and 0.13 when the number of ambiguities is greater than 15. In addition, when the ambiguity search results of the three methods are confirmed, the R-ratio test is combined with the bootstrapping success rate index method. Bootstrapping success rate indicator is set to 99.5%, and the R-ratio test threshold is set to 3.0.

The results of the FAR algorithm are shown in Figure 7, where it can be seen that due to the more complex environment around the mobile station, a large number of floating-point solutions are generated when the FAR algorithm is used for ambiguity fixation. The time series diagram of the number of available co-viewing satellites is given in Figure 8, from which it can be seen that the number of satellites changes more drastically in the first half of the data, indicating that the signals of some satellites may be interfered or blocked by obstacles such as surrounding trees and hillsides during this period. Therefore, the signal quality is affected, causing frequent increases or decreases in the number of available satellites, which in turn affects the overall ambiguity fixation.

The results of the conventional PAR algorithm are shown in Figure 8, from which it can be seen that the number of floating-point solutions is significantly reduced after adopting the partial ambiguity fixation algorithm, especially in the first half of the data. The calendar elements that appear as floating-point solutions in the FAR algorithm are almost all fixed in the PAR algorithm, which verifies the effectiveness of the partial ambiguity fixation algorithm. However, a large number of floating-point solutions still appear in the second half of the data, indicating that the conventional PAR algorithm does not eliminate all satellites that affect the ambiguity solution.

The results of the new algorithm are illustrated in Figure 9, where almost all the solutions are fixed throughout the observation period. Compared to the conventional PAR algorithm, the ephemeris elements that appear as floating-point solutions in the conventional PAR algorithm in the latter half of the data are all fixed in this algorithm, indicating that the algorithm can further identify satellites that have an impact on the blur fixation and thus improve the blur fixation rate.

## 4. Discussion

Table 2 statistically indicates the fixation rates of the results after adopting the three algorithms separately, 34.22%, 71.82%, and 99.89% for the FAR algorithm, the conventional PAR algorithm, and the combined optimization algorithm, respectively. From the statistics, the fixation rate is greatly improved after adopting the partial ambiguity fixation algorithm, which is 37.04% after adopting the conventional PAR algorithm and 65.01% after adopting the new algorithm compared to the FAR algorithm.

The overall accuracy statistics of the solved results of different algorithms are given in Table 3. Combined with Table 2, it can be seen that the presence of floating-point solutions greatly lowers the overall accuracy level of the solved results. After adopting the partial ambiguity fixed algorithm, the accuracy in all aspects is improved regardless of the conventional PAR algorithm or the new algorithm, and the improvement of the new algorithm is especially obvious, and the accuracy of the solved results in the east direction and north direction is within 1 cm after adopting the new algorithm. This further illustrates that the partial ambiguity fixation algorithm can effectively perform quality control on the whole circumference ambiguity solution process in complex environments, thus improving the overall accuracy and fixation rate of the results and reflecting the superiority of the new algorithm.

## 5. Conclusions

In this paper, a fast ambiguity solving method for network RTK reference stations is proposed, which firstly introduces the IGG III model in robust M estimation into the extended Kalman filtering process to achieve the anti-variance effect, and then effectively suppresses the influence of coarse variances on the settlement results, improves the reliability of the solved results, and ensures their accuracy. The floating-point ambiguity is estimated by the robust extended Kalman filtering technique, and the ambiguity fixed subset is preferably selected by using the partial ambiguity solution method; and finally, the ambiguity is fixed by combining LAMBDA with the joint test of R-ratio and bootstrapping success rate index. The method shortens the convergence time of long-baseline ambiguity initialization in network RTK and overcomes the ambiguity fixation success rate problem during satellite lift. Furthermore, compared with FAR and conventional PAR methods, the new method has significantly improved the accuracy in all aspects. This shows that the new algorithm not only reduces the initialization time of the ambiguity solution for network RTK reference stations but also greatly improves the fixation rate of the ambiguity solution for the whole circumference and the accuracy of overall settlement results.

## Figures and Tables

**Figure 1 sensors-22-00165-f001:**
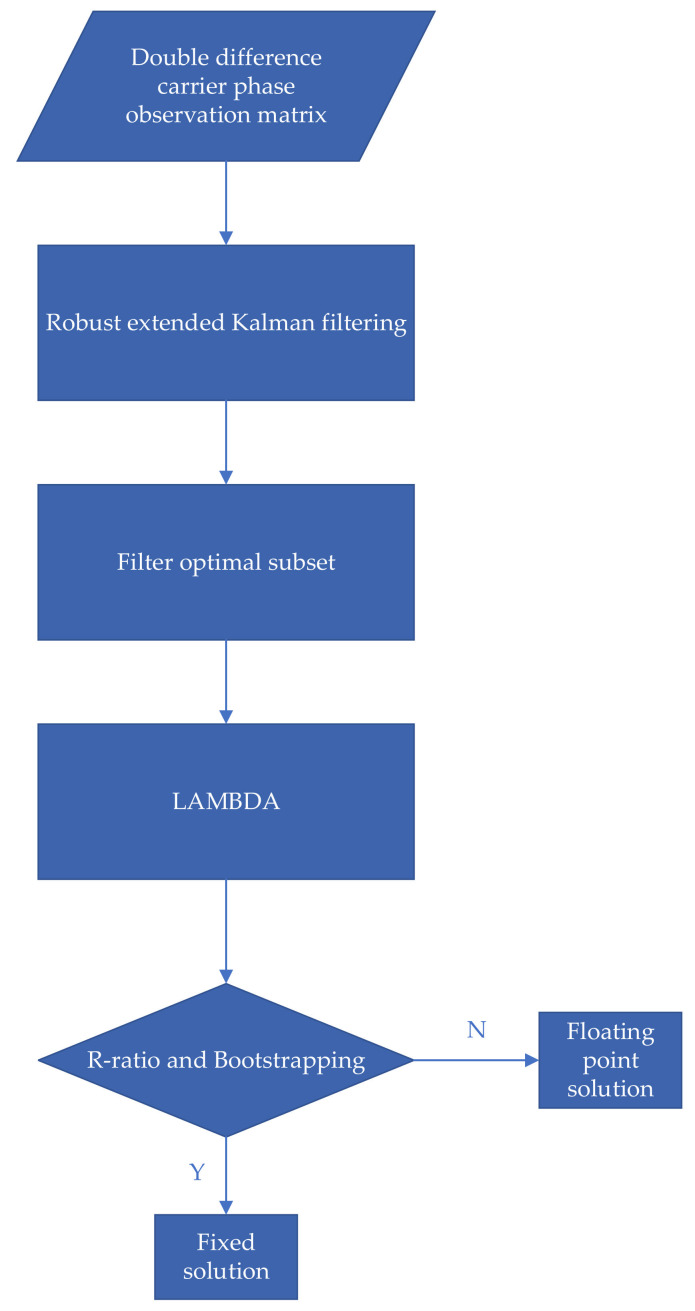
Flow chart of ambiguity resolution steps of network RTK reference station.

**Figure 2 sensors-22-00165-f002:**
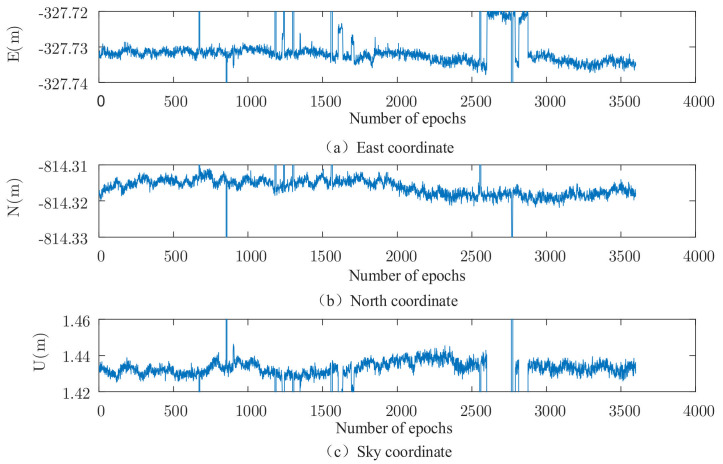
Measured data 1, traditional EKF three-dimensional baseline results.

**Figure 3 sensors-22-00165-f003:**
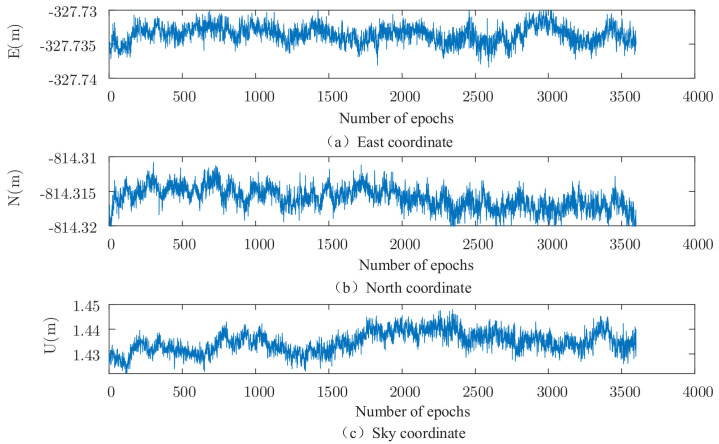
Measured data 1, three-dimensional baseline results of robust EKF.

**Figure 4 sensors-22-00165-f004:**
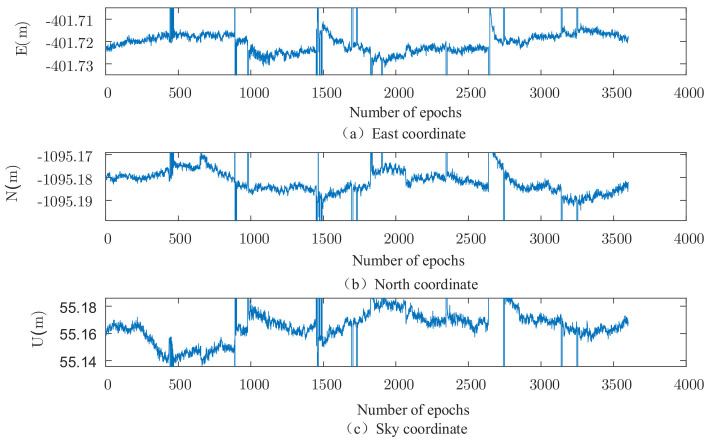
Measured data 2, three-dimensional baseline results of traditional EKF.

**Figure 5 sensors-22-00165-f005:**
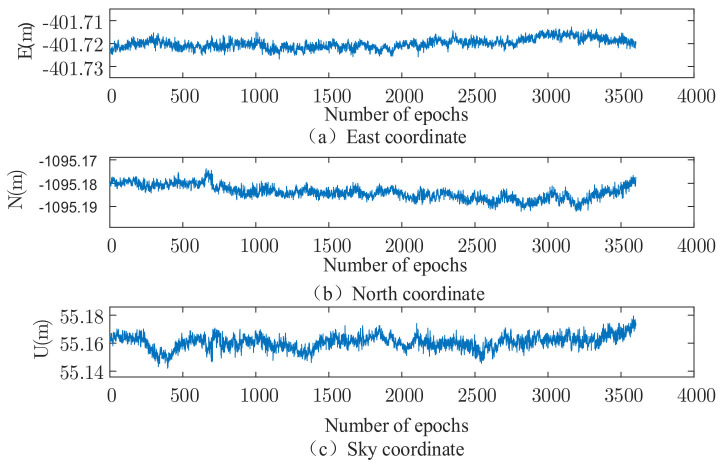
Measured data 2, three-dimensional baseline results of robust EKF.

**Figure 6 sensors-22-00165-f006:**
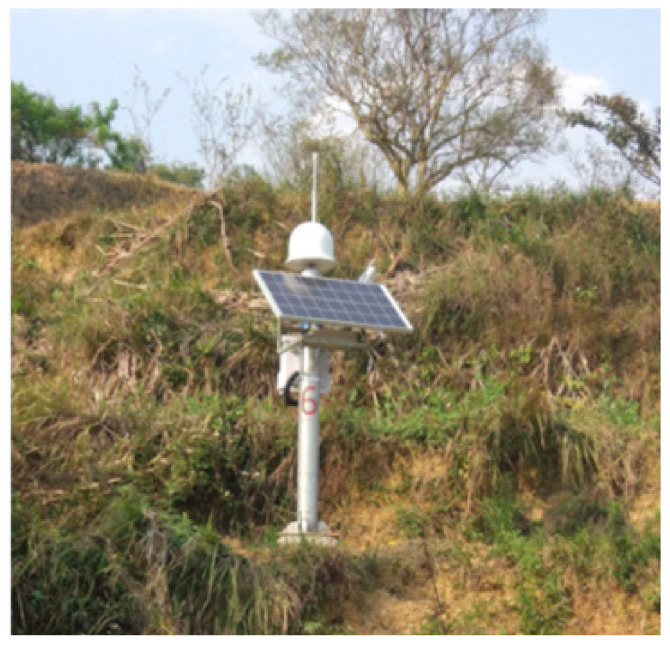
Experimental data collection site.

**Figure 7 sensors-22-00165-f007:**
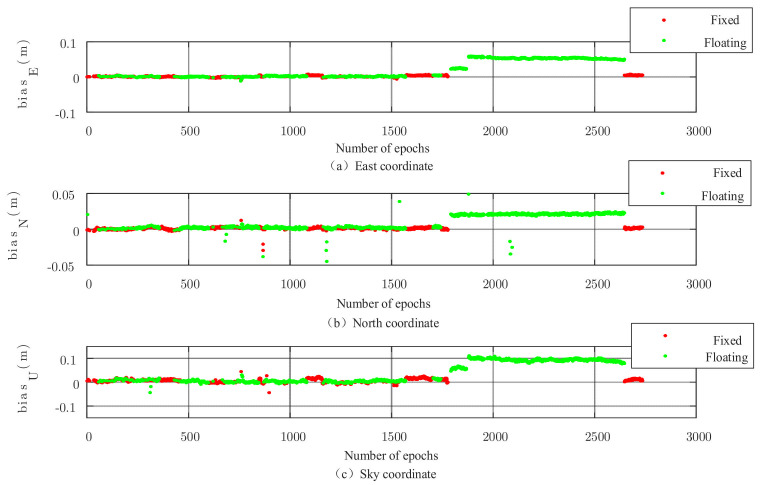
3D baseline solution of the FAR algorithm.

**Figure 8 sensors-22-00165-f008:**
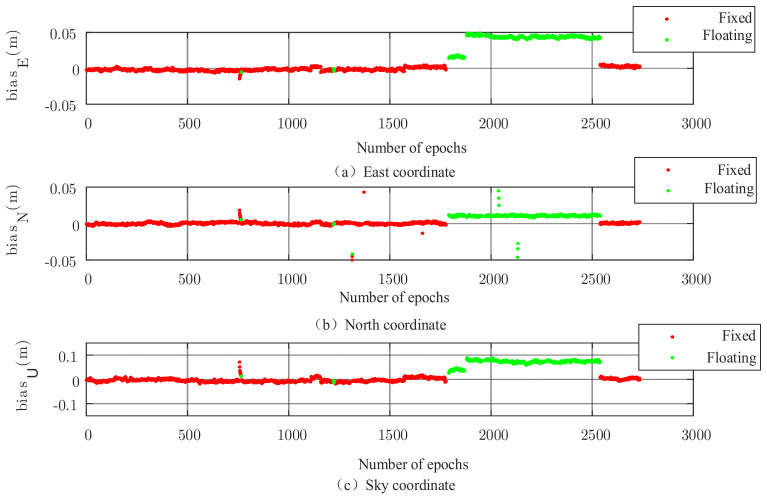
3D baseline results of conventional PAR algorithm.

**Figure 9 sensors-22-00165-f009:**
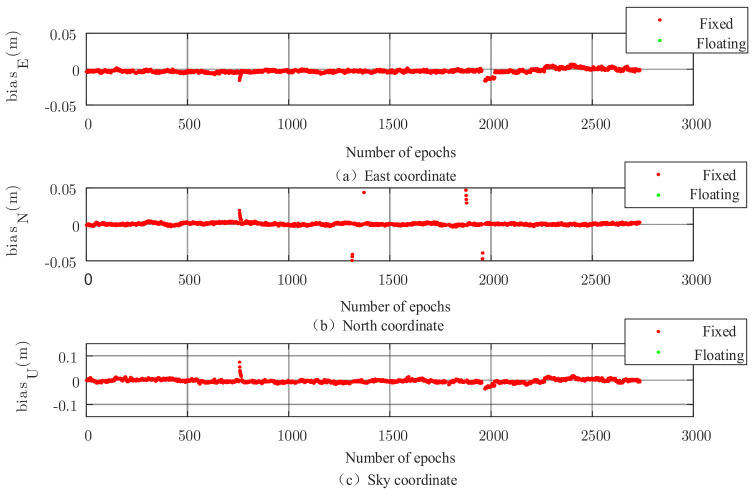
Three-dimensional baseline solution of the combined optimization algorithm.

**Table 1 sensors-22-00165-t001:** Combination optimization algorithm parameter setting.

Related Parameters	Numerical Value
Elevation Threshold	20
Consecutive epochs	10 Epoch
Signal-to-noise ratio	35 dBHz
GDOP	2.0

**Table 2 sensors-22-00165-t002:** Fixed rate statistics of each algorithm.

Algorithm	FAR	PAR	N-PAR
successful epochs	936	1983	273220
failed epochs	1799	751	310 Epoch
Fixed rate	34.22%	71.82%	99.89

**Table 3 sensors-22-00165-t003:** Calculation accuracy statistics of each algorithm.

Algorithm	ΔE	ΔN	ΔU
FAR	0.0601	0.0259	0.0915
PAR	0.0459	0.0151	0.0493
N-PAR	0.0060	0.0089	0.0152

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
