# Peer review of "A Partial Carrier Phase Integer Ambiguity Fixing Algorithm for Combinatorial Optimization between Network RTK Reference Stations"

_sensors, 2021, doi:10.3390/s22010165_

Round 1
Reviewer 1 Report
Dear author,
In „A fixed algorithm of ambiguity among the network RTK reference stations” you present a method for ambiguity fixing for reference stations. This method is only new in a limited detail and this detail in not sufficiently described. The whole text is hard to read and you have to improve your English writing substantially. It also seems that you did not take your time to write the text appropriately, as some paragraphs are in colour, or in a different layout and some sentences are not complete.
Starting with the title the abstract and introduction has to be improved. The title does not tell anything about what is new in your method. Why should someone read your text?
The abstract: The first sentence can be split into two aspects why it is not easy to solve for all ambiguities. Please make two sentences. Show, that this is your motivation to change the existing algorithm. I think that “circumference ambiguity dimension” is not a good expression of what you want to tell us. Please state explicitly what is the problem! What does the phrase “the observation values are easy to have coarse difference values” mean? Please write more scientifically.
Robust not robush this is more often in the text
Define M-estimation
Introduction: First sentence to long and has two different aspects, which do not fit together.
Second sentence: Which responses possesses the advantages you list here
Line 40: precise differential what?
43: real-time what?
47: results appear obvious deviation no English phrase nor scientific sounding
50: what does MW mean?
59: circumferential hopping information?
68: no whole sentence
71: no whole sentence
Define fuzzy degree estimation
86: equation
Paragraph 2.1 : to often: represents,
92: c is the speed of light, which can be determined…
104: a is a vector of integers
Paragraph 2.2: Please compare the usual way of ambiguity fixing maybe also in a flowchart, and highlight what is new in your algorithm. It is totally not clear what is done up to now and what do you change.
147: you propose white noise or you can prove to be white?
150: What does coarse difference in observation data mean?
151: period jump – phase jump?
154: what is M-estimation, what is a anti-divergence Kalman filter, citation?
166: What is the IGG III model?
169 now the layout changes
195: citation for Teunissen is missing
225: what means an azimuth greater than 90°? What is Q1 -4 and how is it related to H1?
244: what is a calendar element?
261: what residual plots?
Conclusion: now it is clear what you have done, but this I have not understood from what is written before! But it stays unclear for me what the IGG III model and M-estimation is!
Best regards
Author Response
Point 1: Line 40: precise differential what?
Respons 1: Precise differential:Precise differential means that there are errors such as satellite clock errors in the positioning process. The reference station calculates these errors by making errors, and then transmits them to the monitoring station in real time to eliminate the errors of the monitoring station. These errors are precise differential.
Point 2: 43: real-time what?
Respons 2: real-time:Real-time means fast solution speed or real-time positioning.
Point 3: 50: what does MW mean?
Respons 3: MW combination method:This is a combined observation algorithm proposed by Melbourne and wubbena.
Point 4: 59: circumferential hopping information? 151: period jump – phase jump?
Respons 4: Circumferential hopping information: Circular hopping information refers to the jump or interruption of the whole week count due to the loss of lock of satellite signal.
Point 5:Define fuzzy degree estimation
Respons 5: Definition of carrier phase integer ambiguity: the conventional method of measuring distance is to multiply time by speed, but the error is relatively large. Therefore, I measure the number of carrier wavelengths from the satellite to the ground receiver. Because the carrier frequency is very high and the wavelength is short, the accuracy is high. The number of carrier wavelengths from the satellite to the ground receiver is called carrier phase integer ambiguity.
Point 6:Paragraph 2.2: Please compare the usual way of ambiguity fixing maybe also in a flowchart, and highlight what is new in your algorithm. It is totally not clear what is done up to now and what do you change.
Respons 6: Paragraph 2.2: In the flow chart, this paper innovates the robust Kalman filter algorithm, which can effectively suppress the influence of gross error on the solution results, and optimizes the partial ambiguity fixing strategy. The conventional partial ambiguity fixing algorithm simply eliminates the satellite according to the satellite elevation. This paper innovates the partial ambiguity fixing algorithm through multi-dimensional optimization.
Point 7: 147: you propose white noise or you can prove to be white?
Respons 7: This paper introduces that when using Kalman filter algorithm, the premise is that the observation signal should meet Gaussian white noise, but in the actual environment, the signal is difficult to meet the properties of Gaussian white noise, so we innovate robust Kalman filter algorithm.
Point 8: 150: What does coarse difference in observation data mean?
Respons 8: coarse difference values:Coarse difference values refer to the observation error with an absolute value greater than three times the mean square error in the received satellite signal under the same observation conditions. For example, there is a sudden large or small observation value in the middle. When there are many obstructions in the observation environment, the observation values are prone to coarse difference values. The influence of coarse difference values is that due to the sudden large or small individual observation values, the filter results diverge, resulting in a large error of solution results.
Point 9: 154: what is M-estimation, what is a anti-divergence Kalman filter, citation?
Respons 9: M-estimation:The full name of M-estimation is called mean estimation M-estimation is a robust estimation method based on least square estimation, which has the advantage of robustness.
anti-divergence Kalman filter:Anti divergence Kalman filter is a new filtering algorithm proposed in this paper, which can effectively suppress the influence of coarse difference values on observations
Point 10: 166: What is the IGG III model?
Respons 10: IGG III model: IGG III model is a model in robust M-estimation, and it is one of the weight factor functions. The adaptive weight reduction factor in the model is introduced into Kalman filter to suppress coarse difference values
Point 11: 225: what means an azimuth greater than 90°? What is Q1 -4 and how is it related to H1?
Respons 11: he azimuth range of the satellite is 0°- 360°. 0°- 90°is Q1, 90°- 180° is Q2, 180°- 270°is q3,270°-360°is Q4, and then one satellite is proposed from Q1, Q2, Q3 and Q4 respectively, and then the remaining satellite set is H1.
Point 12: 244: what is a calendar element?
Respons 12: Calendar element: Calendar element is a unit describing time in satellite positioning, which represents the observation results of satellites at different times.
Point 13: 261: what residual plots?
Respons 13: Residual plots: The residual plots show the graph after processing the coarse difference values with the new algorithm. It can be seen from the graph that the coarse difference values can be effectively suppressed.
Other questions have been revised in the original text and marked in red font.

Reviewer 2 Report
A Fixed Algorithm of Ambiguity Among the Network RTK Reference Stations
Brief summary
The Authors presented the new method of known ambiguity problem of the GNSS positioning in application to the RTK method. They proposed a new method of the ambiguity problem solution based on a partial ambiguity fixation. The test conducted on measurement multiple system data showed a significant improvement of the ambiguity fixation efficiency.
Broad comments
The proposed idea looks interesting and the test results suggest that it is unless prospective but the manuscript content needs significant revision.
The main objection deals with the terms used in the article. The authors use term 'corase difference values' for the measurement results affected by extraordinary errors. In my opinion, the term 'outlier' should be used instead.
Another example of the author's lexical 'creativity' is 'Robush extended Kalman filtering'. The word 'robush' is used many times in the manuscript by sometimes it is used interchangeably with the (proper in my opinion) word 'robust' eg. Section 2.3 title.
Specific comments and minor mistakes
- Line 78: Authors should also define the term 'calendar element'
- Lines 86-93: Why is the text in red?
- Line 144: The word 'Kalman' should start from a capital letter
- Lines 247 and 268: The EKF abbreviation should not be preceded by the word 'extended' (It contains this word)
- Symbols used in Equations 16, 17, and 18 should be explained.
- Lines 224-235: The fragment is not clear. Please rewrite it in a more communicative manner.
- Lines 272-283: This description suits rather section 2 'Materials and Methods'.
- Although all the chart figures have proper resolution, they are rather, small and not legible until zoomed.
There are also quite numerous redactional mistakes concerning fonts, symbol formatting e.t.c.
Author Response
Your suggestion has been revised in the original text and marked in blue font.
Thanks!

Round 2
Reviewer 1 Report
Dear author,
your corrections are not sufficient. Your English has to improve. I marked all I found in the text. The questions I asked are not for my understanding but to help you improving your English. Soanswer them in your text, please.
Best regards

Author Response
Hi,thank you very much for your suggestions. I have revised the original text. Please check it. Thank you again for your suggestions.

Reviewer 2 Report
The manuscript has been changed in many places but there are still many lacks.
With regard to my previous comments, three of them are not sufficiently addressed:
1. There is still a word 'robush' left in many places of the text. Please change it to 'robust'
2. Although the authors introduced the proper term 'gross error' in the Abstract and the Introduction section (line 127), they have still 'coarse error', 'coarse difference', and 'coarse observations' left in the text.
3. Symbols used in Equations 16, 17, and 18 are still not explained.
Author Response
Hi, thank you very much for your suggestions. I have revised the original text. Please check it. Thank you again for your suggestions.

Round 3
Reviewer 1 Report
The manuscript has improved.